# Overall Survival of Patients with Myxofibrosarcomas: An Epidemiological Study

**DOI:** 10.3390/cancers14051102

**Published:** 2022-02-22

**Authors:** Chiel A. J. van der Horst, Sabien L. M. Bongers, Yvonne M. H. Versleijen-Jonkers, Vincent K. Y. Ho, Pètra M. Braam, Uta E. Flucke, Johannes H. W. de Wilt, Ingrid M. E. Desar

**Affiliations:** 1Department of Medical Oncology, Radboud University Medical Centre, 6525 GA Nijmegen, The Netherlands; chiel.vanderHorst@radboudumc.nl (C.A.J.v.d.H.); sabien.bongers@radboudumc.nl (S.L.M.B.); yvonne.versleijen-Jonkers@radboudumc.nl (Y.M.H.V.-J.); 2Department of Research and Development, Netherlands Comprehensive Cancer Organization (IKNL), 3511 DT Utrecht, The Netherlands; v.ho@IKNL.nl; 3Department of Radiation Oncology, Radboud University Medical Centre, 6525 GA Nijmegen, The Netherlands; p.braam@radboudumc.nl; 4Department of Solid Tumours, Princess Máxima Centre for Pediatric Oncology, 3584 CS Utrecht, The Netherlands; uta.flucke@radboudumc.nl; 5Department of Pathology, Radboud University Medical Centre, 6525 GA Nijmegen, The Netherlands; 6Department of Surgery, Radboud University Medical Centre, 6525 GA Nijmegen, The Netherlands; hans.dewilt@radboudumc.nl

**Keywords:** myxofibrosarcoma, local recurrences, metastasis, prognostic factors, epidemiology

## Abstract

**Simple Summary:**

Myxofibrosarcoma (MFS) is a rare soft tissue sarcoma type with high local recurrence and amputation rates. Robust epidemiological data on overall survival of patients with MFS and prognostic factors are lacking due to the rareness of the disease. In this study, we therefore report prognostic factors and real-life outcomes of the largest myxofibrosarcoma cohort to date including 908 patients. Five-year overall survival was 68%. Multivariate analyses revealed known prognostic factors for OS, such as age, tumour size, and histological grade with the addition of sex. In a more detailed subcohort of 177 patients, 39% developed a local recurrence and 28% distant metastases. The survival outcomes and recurrence rates found in this study emphasize the need to improve treatment strategies.

**Abstract:**

Myxofibrosarcoma (MFS) is a rare mesenchymal soft tissue sarcoma type, with a high local recurrence (LR) rate. Robust epidemiological data on MFS are lacking. We, therefore, aimed to identify prognostic factors and describe real-life outcomes of a large cohort of 908 MFS patients obtained from the nationwide database of the Netherlands Cancer Registry and diagnosed between 2002 and 2019. Median Overall survival (OS) was 155 (range 0.1–215) months, with a five-year OS of 67.7%. No improvement of OS was found over time. Multivariable Cox regression survival analysis demonstrated known prognostic factors for OS, such as older age, tumour size, and histological grade with the addition of sex. Surgery at sarcoma expertise centres, instead of general hospitals, was associated with better OS outcomes. In a subcohort of 177 patients, 39% developed LR with a median time to recurrence of 20 months. From LR on, the median OS was 64.0 months (CI 95% 38.5–89.5). In 28%, distant metastases were diagnosed with a median OS of 34.3 months (CI 95% 28.8–39.8) after diagnosis of the primary tumour. In this largest nationwide cohort so far, survival outcomes and recurrence rates for MFS patients did not improve over time, emphasizing the need to improve treatment strategies and suggesting a role for sarcoma expertise centres.

## 1. Introduction

Myxofibrosarcoma (MFS) is a rare mesenchymal malignancy, most commonly arising in the extremities as a slow-growing painless mass [1]. MFS accounts for 5% of all soft tissue sarcomas (STS) with STS showing an annual incidence of approximately 800 cases in the Netherlands [2,3]. The World Health Organization (WHO) classified MFS as a separate diagnostic entity in 2002, whereas previously MFS was classified as malignant fibrous histiocytoma. The adjustment in the classification of MFS was required since the myxoid variant of malignant fibrous histiocytoma could be classified as a distinct entity [4]. 

Robust epidemiological data on MFS are lacking due to the rareness of the disease and the renaming by the WHO in 2002. So far, only a few studies with small patient numbers are available, reporting a five-year overall survival (OS) between 61% and 77% [5,6,7,8]. To date, known prognostic factors for OS in MFS include older age, tumour size, histological grade, and surgical margins [5,6,7,8]. 

Compared to other STS, MFS is clinically known to have high local recurrence (LR) rates, ranging between 16% and 61% compared to around 10% in other types of STS [1,5,8,9,10,11,12,13,14,15]. This can be explained by the infiltrative growth pattern of MFS into the surrounding connective tissues, which makes identification of the true size of the tumour during surgery challenging [6,16,17,18]. As a result, amputations are more frequently performed in MFS compared to other STS types (17–41% vs. 5%) [19,20,21]. In addition to surgical margins, older age is also reported as risk factor for local recurrences [6,8,9,10,22].

Distant metastases rates reported in the literature range between 15% and 38% [1,5,6,7,9,14,15]. Reported risk factors of distant metastasis include histological grade and tumour size [6,9]. Once metastasized, no MFS specific treatment strategies are available and prognosis is poor [5].

We aimed to expand the limited amount of epidemiological data on MFS by describing the patient and tumour characteristics and clinical outcomes from a large Dutch cohort of MFS patients diagnosed between 2002 and 2019, using data from the Netherlands Cancer Registry (NCR). 

## 2. Materials and Methods

### 2.1. Patient Data

For this retrospective cohort study, we obtained the clinical data of patients with MFS by using a nationwide database from the Netherlands Cancer Registry (NCR). Regarding recurrences the NCR only reported follow-up of patients diagnosed from 2007 until 2011. This cohort of patients includes Dutch patients with histologically confirmed MFS from 1 January 2002 until 31 December 2018, using the ICD-O-3 histology code 8811/3 for MFS. For the years 2002–2012, this means that also the myxoinflammatory fibroblastic sarcomas (MIFS) and the low grade fibromyxoid sarcomas were included. The database from the NCR consists of extensive information on patient, tumour, and treatment characteristics, including data about the type of hospital where patients were diagnosed and treated. The information in the database is collected from the hospital records by trained and dedicated data managers after notification of the pathology laboratories. Follow-up information about the vital status is obtained via linkage with the local reports from the Dutch Municipals Records. The grading classification from the FNCLCC (Fédération Nationale des Centres de Lutte Contre le Cancer) is used for the grading of the tumours. Surgical margins after tumour resection are classified into four different resection levels, namely R0 (negative/clear margins), R1 (positive margins, microscopic), R2 (positive margins, macroscopic), and RX (surgery margin unknown). The primary objective is to determine median OS. As part of a pilot of the National Cancer Registry, a more detailed dataset, including a follow-up period, has been collected in all MFS patients diagnosed in the period between 2007 and 2011. Secondary objectives in this subcohort include recurrence rate, prognostic factors, the influence of being treated in a sarcoma expertise centre and time (2002–2010 vs. 2010–2018) on clinical outcomes. To differentiate between sarcoma expertise centres and non-sarcoma centres we used the list from the Dutch Patient Association for Sarcomas [23]. Local recurrence was defined as recurrence of MFS within the scar of the primary treatment and locoregional recurrence as recurrence of MFS within the same body part and without hematogenic spread (e.g., skip lesions). 

### 2.2. Statistical Analysis

Analysis of variance (ANOVA) tests were used to examine the differences in various patient and disease characteristics for continuous variables. For the categorical variables, the chi-squared test was used. Descriptive analysis of the survival estimation, including OS, was based on Kaplan–Meier models, where OS was calculated starting at the date of definitive diagnosis until date of death. The log-rank test was used to compare differences in survival between different groups. Cox proportional hazard models were used to identify independent prognostic factors influencing overall survival, local recurrence free survival or distant metastasis free survival. Factors used in the Cox proportional hazard model were selected when significant in the univariable tests (*p* < 0.05).

Information missing in the database was coded as ‘unknown’. Multiple imputation was used to include patients with missing data in the analyses. We used the Bayesian regression method with Predictive Mean Matching for scale variables. We generated 5 imputed datasets which were made using all prognostic variables. Before starting the multiple imputation procedure, a starting point was set with the random number generator at a fixed value of 950. The statistical analyses were performed using IBM SPSS statistics, version 26. A *p*-value of < 0.05 was considered statistically significant.

## 3. Results

### 3.1. Patient and Clinical Characteristics

Between 2002 and 2018 a total of 910 cases with a reported MFS diagnosis were identified in the NCR. Two patients were excluded because the vital status was not reported at follow-up in the database. Median age at diagnosis was 67 years (range 18–98) and 54% of the patients were male (492/908). The majority of patients had no record of previous cancer in their medical history (85%, 766/908). The tumour size was >5 cm in 498 patients (55%) and ≤5 cm in 375 patients (41%). In 35 patients (4%), the size was unknown. Most tumours were superficially located (46%, 420/908), 267 deep (29%), and unknown in the rest of the patients (24%, 221/908). The extremities were the most common primary localization (76%, 686/908), followed by the trunk (20%, 183/908), head and neck (23%, 24/908), (retro)peritoneum (0.55%, 5/908), and mediastinum/heart/pleurae (0.4%, 4/908). In five patients, the tumour was located in a specific organ (0.55%), and in one patient the location was unknown (0.1%). Histological grade III was most common (39%, 358/908), while grade II was found in 21% (188/908) and grade I in 27% (241/908). In 13%, data on the histological grade were lacking (121/908). In 42 cases, patients were reported with distant metastases at diagnosis (5%).

For primary treatment, the majority of patients underwent surgery (92%, 837/908), resulting in no residual disease in 64% (539/837). R1 resections were reported in 20% (164/837), R2 resections in 2% (13/837), and unknown in 14% (121/837). Additional treatment with radiotherapy was given in 498 patients (55%), including 188 patients receiving neoadjuvant radiotherapy (21%), 291 patients receiving adjuvant radiotherapy (32%), and 19 patients receiving both neoadjuvant and adjuvant radiotherapy (2%). In 28 patients (3%) definitive radiotherapy was the primary treatment. Few patients underwent (neo)adjuvant systemic therapy (2%, 15/908). 

### 3.2. Overall Survival

Median follow-up was 55.6 months (range 1–215 months). Median OS for the entire cohort was 155 months (CI 95% 126–184 months). One-year, five-year, and ten-year overall survival rates were 91.5%, 67.7%, and 54.4%, respectively. Comparing the OS of patients with MFS diagnosed in 2011–2019 with patients diagnosed in 2002–2010 showed no significant difference (*p* = 0.2) (Figure 1).

### 3.3. Prognostic Factors

Univariable analyses show that the factors age ≥65 years, male sex, cancer in medical history, tumour size >5 cm, deep tumour depth, high histological grade, distant metastases, no surgery, and R1 or R2 residual disease, correlated with poorer OS (sex *p* = 0.03, other variables *p* < 0.01) (Appendix A). Neoadjuvant and/or adjuvant radiotherapy was correlated with better OS (*p* < 0.01). 

In addition, we investigated whether surgery in a dedicated sarcoma expertise centre did influence the patient outcomes compared to non-sarcoma specialized hospitals. OS was significantly improved when the surgery was performed in a sarcoma expertise centre rather than in a non-sarcoma centre (*p* = 0.02), while the centre of diagnosis did not influence OS outcomes (Appendix A). Median OS for patients with surgery in a non-sarcoma centre was 126.0 months (CI 95% 95.1–156.8), whereas OS for patients with surgery at a sarcoma centre was 156.8 months (CI 95% 135.4–178.2). However, this effect was not confirmed in the multivariate analysis in which we only found a trend (Table 1).

Multivariable analyses show that patients ≥65 years have a worse OS than patients <65 years (HR 2.7, 95% CI 2.0–3.6, *p* < 0.01) (Table 1). Sex, cancer in medical history, tumour size, histological grade, distant metastases, surgery, and residual disease are independent prognostic factors. Adjuvant radiotherapy was an independent prognostic factor associated with better OS (HR 0.6 CI 95% (0.4–0.8, *p* ≤ 0.01).

### 3.4. Recurrence and Distant Metastases

More detailed follow-up data on the development of local recurrences and distant metastases were available for 177 patients (Table 2, Figure 2 and Appendix A). Of these 177 patients, 69 patients (39%) developed a local recurrence, with a median time to recurrence of 20 months (range 1.7–88.5 months). A locoregional recurrence was found in six patients (3%). Out of the 177 patients, 50 patients developed distant metastases (28%), with a median time to metastases of 15.3 months (range 3.8–155.0 months). There were 19 patients with both local recurrences and distant metastases, OS for patients with a local, regional, or distant metastasis was significantly lower than for patients who did not have a recurrence or metastasis (*p*-value local recurrence < 0.01, regional recurrence < 0.015 and distant metastasis < 0.01). 

Factors associated with local recurrence free survival and metastasis free survival are presented in Appendix A. 

## 4. Discussion

In this study, we described the survival outcomes and prognostic factors of the largest MFS specific patient cohort so far, containing 908 patients. 

The overall median survival of patients with MFS was 155 months, with a five-year survival rate of 67.7%. This five-year OS is within the 61–77% range reported previously in a total of four studies reporting on 433 patients in total [5,6,7,8].

Multivariable analysis showed that there are various patient and clinical characteristics that influence survival in patients with MFS. To date several studies found older age, tumour size, histological grade, and distant metastases at diagnosis to be predictors of OS [5,6,7,8]. In our study, we confirmed these associations. Tumour depth on the other hand was described in the literature to have no effect on survival and in this study we confirm this [5]. 

In contrast to the other prognostic factors found, sex differences have not been identified as a prognostic factor yet. On top of males having a slightly higher incidence of MFS, we also found males to have a significant lower OS than females with MFS. In recent studies, only Kaya et al. mentioned a non-significant increase in local recurrence-free survival in female patients [10]. The difference can possibly be explained by male cancer patients having a shorter survival compared to women in general, however this difference tends to be negligible in the elderly [24]. Biologically there is no known explanation for the difference in OS in MFS patients. As sex is a non-influenceable factor, the found impact on survival can still help in estimating the prognosis and optimal treatment plan.

Surgery is the main treatment option for MFS and the importance of clear margins has been described multiple times in the literature [6,8,9,20,22,25,26]. In this study, we confirm the importance of negative margins on survival outcomes. The literature on the value of (neo)adjuvant radiotherapy in addition to surgery on limited numbers of MFS patients is conflicting [6,8,9,21]. In this retrospective cohort approximately 20% of patients received neoadjuvant radiotherapy while 30% received adjuvant radiotherapy. Only adjuvant radiotherapy was directly associated with both better local recurrence free survival and OS. This is in contrast to clinical practice, where neoadjuvant radiotherapy is increasingly preferred above adjuvant radiotherapy since smaller fields are needed with less morbidity and might, therefore, reflect a selection bias in this retrospective cohort. 

Within the 177 patients whose recurrence status was registered, after 5 years, 37% of the patients developed an LR rate and 26% distant metastases. These findings are in line with previous publications reporting local recurrence rates between 16% and 61%, and distant metastases rates between 15% and 38% [1,5,6,7,8,9,10,11]. As expected the local recurrence rate was higher than that of STS in general which is around 10% [12].

Despite all advances in diagnostic opportunities, radiotherapy techniques and systemic treatments, we could not demonstrate an improved overall survival of MFS patients over the past 20 years. This is in contrast to a recent report of the NCR in all types high grade STS, showing an improved three-year survival from 50% to 61% compared to a cohort from 10 years ago [27]. A way to improve clinical outcomes of MFS patients might be centralization of care. In our study we found a trend towards an increase in overall survival when surgery is performed in a sarcoma expertise centre. This is consistent with other publications. Vos et al. found that surgery of low-grade and deep-seated STS in a high-volume centre (≥20 resections annually) improved survival when compared to patients who had surgery in a low-volume hospital (1–9 resections) [28]. Kikuta et al. found similar results in a cohort of 100 MFS patients and marked primary unplanned resection at non-specialized facilities as a risk factor related to poor prognosis [26]. Additionally, Blay et al. showed that surgery of STS and visceral sarcomas in a sarcoma expertise centre positively-correlated with local relapse free survival and relapse free survival [29].

Other future possibilities to improve survival outcomes of MFS not examined in this study but worth mentioning are in depth insights in molecular drivers of MFS, insights in the immune micro-environment and advanced imaging techniques possibly even during primary surgery to increase the R0 resection rates. To better understand which prognostic and predictive factors determine survival outcomes in MFS, future case–control studies or prospective cohort studies collecting more detailed information on for example social and environmental exposures, as well as radiomics and genomics are recommended.

Our study has several limitations. First, patients were selected based on ICD-O-3 codes within the Netherlands Cancer Registry without central histology revision. The histopathological diagnosis of MFS might be challenging for pathologists not dedicated to sarcomas. Furthermore, within the registry also myxoinflammatory fibroblastic sarcoma and low-grade fibromyxoid sarcomas are part of the ICD category and, therefore, included in the database. These diseases are extremely rare, with an incidence of low-grade fibromyxoid sarcomas of only 0.6% of all STS and for myxoinflammatory fibroblastic sarcoma only 250 cases described in literature until 2013 [30,31]. Second, missing data might influence the outcomes of the multivariable analyses. Therefore, we used multiple imputation. Third, the NCR is coupled to the Dutch municipal reports for its survival data. This means that general overall survival data provided are not disease specific. More detailed follow-up data were only available in a subcohort of 177 patients.

## 5. Conclusions

In conclusion, in this largest MFS cohort so far, no improvement in OS was found and high local recurrence rates are confirmed. Sex is added as prognostic factor. Surgery within a sarcoma expertise centre might improve survival. 

## Figures and Tables

**Figure 1 cancers-14-01102-f001:**
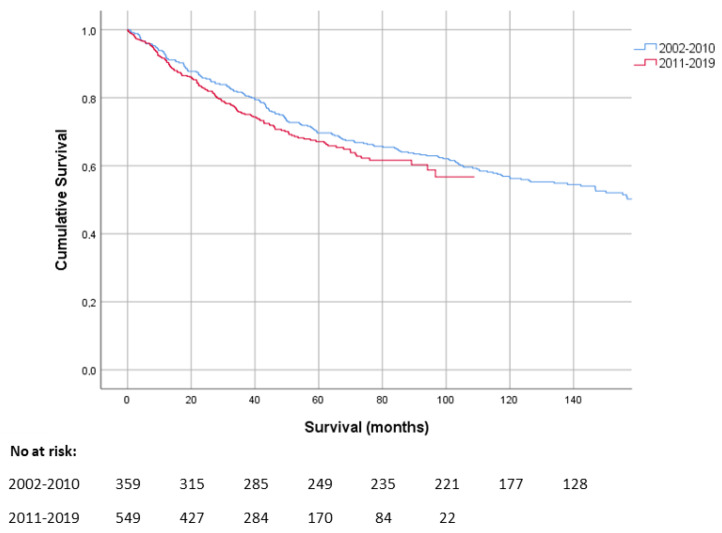
Kaplan–Meier curve showing the survival of MFS patients diagnosed in 2002–2010 in comparison to patients that were diagnosed in 2011–2019.

**Figure 2 cancers-14-01102-f002:**
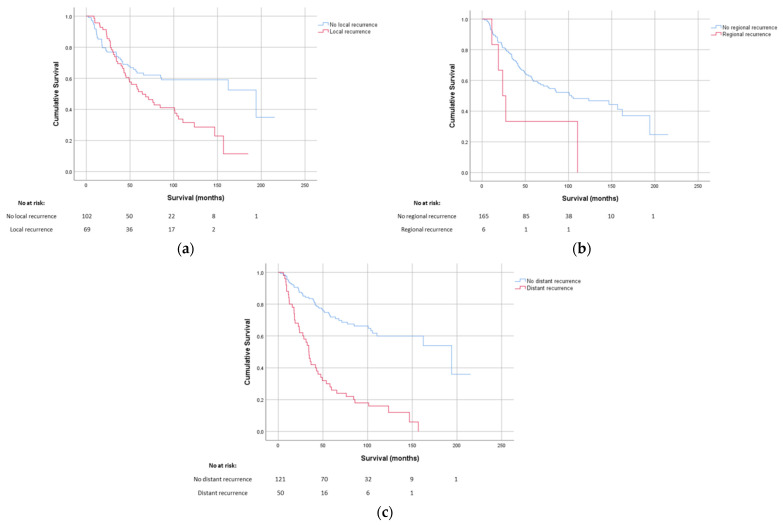
Overall survival of MFS patients with recurrence/metastasis as calculated from date of first diagnosis. (**a**) Kaplan–Meier curves showing the difference in survival with and without local recurrences, (**b**) Kaplan–Meier curves showing the difference in survival with and without regional recurrences, (**c**) Kaplan–Meier curves showing the difference in survival with and without distant recurrences.

**Table 1 cancers-14-01102-t001:** Multivariable survival analysis with imputed data.

Variable	Overall Survival	*p*-Value ^2^
	Hazard Ratio (95% CI) ^1^	
**Age**		
<65 years	Ref.	
≥65 years	2.7 (2.0–3.6)	<0.01 ^3^
**Sex**		
Male	Ref.	
Female	0.7 (0.6–0.9)	<0.01 ^3^
**Cancer in medical history**		
No	Ref.	
Yes	1.5 (1.2–2.1)	<0.01 ^3^
**Tumour size**		
≤5 cm	Ref.	
>5 cm	2.0 (1.5–2.7)	<0.01 ^3^
**Tumour depth**		
Superficial	Ref.	
Deep	1.3 (0.9–1.8)	0.10
**Histological grade**		
I	Ref.	
II	2.3 (1.5–3.8)	<0.01 ^3^
III	3.1 (1.8–5.2)	<0.01 ^3^
**Distant metastases at diagnosis**		
No	Ref.	
Yes	2.5 (1.4–4.7)	<0.01 ^3^
**Surgery**		
No	Ref.	
Yes	0.2 (0.1–0.4)	<0.01 ^3^
**Residual disease**		
R0	Ref.	
R1/R2	1.7 (1.2–2.4)	<0.01 ^3^
**Radiotherapy**		
No	Ref.	
Neoadjuvant	0.8 (0.6–1.1)	0.20
Adjuvant	0.6 (0.4–0.8)	<0.01 ^3^
Neoadjuvant + adjuvant	0.1 (0.02–0.9)	0.04 ^3^
Primary	0.4 (0.2–0.8)	0.02 ^3^
**Surgery**		
Sarcoma centre	Ref	
Other	1.3 (1.0–1.6)	0.087

^1^ CI (confidence interval) ^2^ log-rank test ^3^
*p* < 0.05.

**Table 2 cancers-14-01102-t002:** Recurrence time and effect of recurrences and distant metastasis on overall survival.

Recurrence or Metastases	No (%)	Median Time to Recurrence (Range)	Median OS (Months. 95% CI) ^1^	*p*-Value ^2^
Local recurrence				<0.01 ^3^
Yes	69 (39)	20.0 (1.7–88.5)	64.0 (38.5–89.5)	
No	108 (61)		194.1 (60.3–327.8)	
Regional recurrence				<0.015 ^3^
Yes	6 (3)	17.4 (3.8–82.4)	23.8 (13.4–34.2)	
No	171 (97)		103.5 (51.9–155.2)	
Distant metastases				<0.01 ^3^
Yes	50 (28)	15.3 (3.8–155.0)	34.3(28.8–39.8)	
No	127 (72)		194.1 (88.4–299.7)	

^1^ OS (overall survival) after recurrence/metastasis, CI (confidence interval), ^2^ log-rank test ^3^
*p* < 0.05.

## Data Availability

The data presented in this study are available on request from the corresponding authors.

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
