# Peer review of "Overall Survival of Patients with Myxofibrosarcomas: An Epidemiological Study"

_cancers, 2022, doi:10.3390/cancers14051102_

Round 1

Reviewer 1 Report

To Authors’: The purpose of this study was to describe the patient and tumour characteristics and clinical outcomes of Myxofibrosarcoma (MFS)- a rare mesenchymal malignancy – with patients diagnosed between 2002-2019, and from the Netherlands Cancer Registry. This study is important because MFS is a relatively ‘new’ cancer in that it was formally subclassified from soft-tissue sarcomas in 2002. Thus, few epidemiologic studies have been conducted to date. Despite the importance, the study could be improved by adding detail to the statistical analysis as well as guiding future studies based on these results or results of other soft tissue sarcoma studies. Specific comments follow.

Simple Summary

  1. Abbreviate Myxofibrosarcoma in parentheses before using ‘MFS’ in text.
  2. Should “In this study we therefore report prognostic factors and real life outcomes of the largest myxofibrosarcoma cohort to data … ” be “In this study we therefore report prognostic factors and real life outcomes of the largest myxofibrosarcoma cohort to date …”?
  3. Consider “Five year overall survival was 68%.” In place of “Five year overall survival was found to be 68%.”

Abstract

  1. Include years of diagnosis in the abstract

Background/Introduction

  1. Paragraph 2 (and throughout Background and Discussion): when referring to ‘age’ as a risk factor, consider saying ‘older age’.

Materials and Methods

  1. Clinical data: Consider a more accurate subheader as sex or gender is typically not considered a clinical factor
  2. Clinical data: Please identify in this paragraph whether ‘gender’ or ‘sex’ is the patient factor analyzed and mentioned in Methods, Results, and Discussion. If self-reported gender-identity (cis-gender man, cis-gender woman, trans-gender man, non-binary, etc.) then mention the categories offered for patients to self-identify. If ‘sex’ (male, female, intersex) is asked/recorded from medical/birth record then mention these categories and refer to ‘sex’ throughout. As of now, ‘sex’ appears to be the variable, but ‘gender’ is referenced in text.
  3. Statistical Analysis: Consider “Analysis of variance (ANOVA) tests were” in place of “The variance (ANOVA) test was”
  4. Statistical Analysis: How was the proportional hazards assumption tested?
  5. Statistical Analysis: How were multivariate models built and covariates selected; forward/backward selection, those covariates with p<0.05 in univariate tests, pervious literature, etc.?

Results

  1. Figures 1 & 2: With such small numbers, it is assumed that the KM curves have large standard errors. As such, consider including the cancer cases at risk at periodic time intervals at the bottom of each KM plot? Alternatively, these data can be listed in the supplement and readers should be referred there to know how many people are still be followed at each time point.

Discussion

  1. Paragraph 4, last sentence: Clarify whether ‘sex’ is meant here “As gender is a non-influenceable factor, the found impact on survival can still help in estimating the prognosis and optimal treatment plan.” If truly “gender”, remove ‘non-influenceable ’ as this gender can be modified.
  2. Paragraph 9, first sentence: edit “ICD-O-Q3”.
  3. Paragraph 10, last sentence. Ensure “Gender” or “sex”
  4. As this study had very limited covariates, please discuss what a future case-control or cohort studies should collect to better understand MFS; geographic factors for environmental or social exposures, genomics to understand family history links, etc.?

Reviewer 2 Report

You have to be commended for analysing one of the hitherto largest cohorts of patients with myxofibrosarcoma. The sheer size of the cohort allows for meaninful conclusions on the prognosis of the disease with regard to tumor and treatment characteristics. The analysis and the presentation of results could be improved in some aspects:

In the abstract you write "Median Overall survival (OS) was 155 (range 0.1-215) months, with a five-year OS of 67.7% ", to add some sentences later that "survival outcomes and recurrence rates for MFS patients remained disappointing". While it seems true that survival has not improved with a relevant magnitude during the last 15-20 years, the absolute survival figures do not seem too disappointing for a highly malignant disease. Thus, I suggest re-phrasing the pertinent sentence. 

In the multivariable analysis, I recommend for the variable "surgery" - "yes"/"no" defining "yes" as the ref category, so for all variables, you have the "good" category (e.g. like for R0, sarcoma center etc.) as reference. This makes the results easier to read and appreciate. I also suggest to add the number of patients analysed in each category of the single variables.

"More detailed follow-up data on the development of local recurrences and distant metastases were available for 177 patients": Were these 177 patients a random selection of all patients, or could any selection bias be in place compared to patients not followed up for recurrence? Please elaborate. 

How is local and regional recurrence exactly defined and what are the differences, if any?

Figure 2: "Overall survival of MFS patients after recurrence/metastasis." This legend is misleading. I assume the survival curves show overall survival calculated from first diagnosis, and not from diagnosis of recurrence. Otherwise, how would you even construct the curves for patients without recurrence? Or, if the curves for patients with recurrence start at the date of diagnosis of recurrence, and the curves for patients without recurrence at the date of first diagnosis, the comparison would be totaly skewed.

All survival curves should contain the number of patients at risk at the single time points.
